# Intersection Between Eco-Anxiety and Lexical Labels: A Study on Mental Health in Spanish-Language Digital Media

**DOI:** 10.3390/bs15081102

**Published:** 2025-08-14

**Authors:** Alicia Figueroa-Barra, David Guerrero-Mardones, Camila Vargas-Castillo, Luis Millalonco-Martínez, Angel Roco-Videla, Emmanuel Méndez, Sergio Flores-Carrasco

**Affiliations:** 1Departamento de Psiquiatría y Salud Mental Sur, Facultad de Medicina, Universidad de Chile, Santiago 8900000, Chile; david.guerrero@ug.uchile.cl (D.G.-M.); camila.vargasc@alumnos.uv.cl (C.V.-C.); millaloncoluis@gmail.com (L.M.-M.); eomendez@uc.cl (E.M.); 2Departamento de Lengua y Literatura, Facultad de Filosofía y Humanidades, Universidad Alberto Hurtado, Santiago 8320000, Chile; 3Facultad de Ingeniería, Universidad Católica de la Santísima Concepción, Concepción 4030000, Chile; aroco@ucsc.cl; 4Vicerrectoría de Investigación e Innovación, Universidad Arturo Prat, Santiago 8320000, Chile; 5Facultad de Ciencias de la Salud, Universidad Autónoma de Chile, Santiago 7500000, Chile

**Keywords:** eco-anxiety, solastalgia, climate change, climate anxiety

## Abstract

Background: Eco-anxiety and solastalgia are psychological responses to environmental degradation and climate change. This study examines how these concepts are represented in Spanish-language digital media, considering both emotional dimensions and the profiles of content producers. Methods: We conducted an inductive qualitative content analysis of 120 Spanish-language items (online news articles and selected posts from digital platforms) published between October 2023 and March 2024. Items were identified using a Boolean search strategy and initially filtered by LIWC to detect high emotional-and-anxiety term density; final coding followed grounded-theory procedures, resulting in four thematic categories. Results: The most frequent theme was environmental activism (41%), followed by catastrophic thinking (29%), coping strategies (25%), and loss of meaningful places (6%). Among content producers, citizen participants represented 40%, youth activists 25%, and scientists 15%. Digital media function both as sources of anxiety-inducing content and as spaces for awareness-raising and support. Conclusions: While eco-anxiety is not a clinical diagnosis, it exerts a significant psychological impact—particularly on youth and vulnerable groups. Spanish-language digital platforms play an ambivalent role, amplifying distress yet enabling resilience and collective action. Future interventions should leverage these channels to foster environmental awareness, emotional resilience, and civic engagement.

## 1. Introduction

According to the [31] ([31]), there is a global increase in the number of people living with mental health disorders, reaching 970 million worldwide in 2019. Mental health issues are considered “the great epidemic of the 21st century” ([28]), being conditions that can cause challenges across all areas of life, generating a reduction in both quality of life and healthy life expectancy due to disability, with relevant economic implications for affected individuals and global health systems ([23]; [30]; [29]).

Human health is a multifaceted phenomenon influenced by biological, environmental, social, and economic factors. Biological factors include genetics and the immune system, while environmental factors cover air and water quality, toxin exposure, and climate conditions. Social determinants, such as access to education and healthcare services, along with economic factors, like income levels, significantly impact health ([5]; [26]; [18]; [20]; [6]; [4]).

The complex, multifactorial nature of mental disorders, with environmental influence gaining importance in recent decades, is noteworthy when considering their etiology. Climate change and extreme weather phenomena have emerged as significant risks to mental health, with implications including post-traumatic stress, anxiety, and depression ([4]). Additionally, the water crisis and biodiversity loss, exacerbated by deforestation and natural resource exploitation, increase disease risk and jeopardize food security ([20]). These combined effects form substantial challenges to global public health, interacting complexly to influence the health and well-being of people and communities ([6]; [8]).

Eco-anxiety, a term coined in recent years, refers to a state of deep psychological distress characterized by persistent fear and exacerbated concern over the current and future consequences of the climate crisis. This emotional reaction, identified by Clayton and colleagues ([8]) as an adaptive response to an existential threat, manifests particularly among the youth and vulnerable communities, who feel notably exposed and unprotected against environmental impacts. [24] ([24]) offers a further perspective, interpreting eco-anxiety not only as an emotional response but also as an indicator of a deep existential connection with the planet. Moreover, eco-anxiety can be approached not only from a psychological perspective, focusing on emotional and cognitive responses, but also from a geographical standpoint, where it relates to the future image of places and the perceived deterioration of familiar environments ([27]).

[8] ([8]) and [7] ([7]), in their report for the American Psychological Association, delve into the psychological implications of eco-anxiety. They emphasize that the helplessness and loss of control feelings associated with this condition can trigger a wide range of mental disorders, including generalized anxiety, major depression, post-traumatic stress disorder, and even suicidal ideation. These psychopathological effects are particularly pronounced in populations exposed to extreme weather events or chronic environmental changes. Furthermore, the report underscores social inequalities in experiencing eco-anxiety, highlighting that women, children, marginalized communities, and indigenous peoples are disproportionately impacted by the psychosocial impacts of climate change, exacerbating pre-existing inequities in mental health. In this context, eco-anxiety not only reflects the psychological consequences of the environmental crisis but also poses a challenge to collective mental health and social equity.

The concept of solastalgia, introduced by Australian philosopher Glenn [1] ([1]), complements eco-anxiety. This neologism, combining Latin “sōlācium” (comfort) and Greek “-algia” (pain), refers to the psychological distress experienced due to environmental degradation in a place previously considered familiar and comforting. Unlike nostalgia, which involves longing for a distant past, solastalgia refers to the pain of witnessing the degradation of one’s environment while living within it.

[2] ([2]) illustrate this concept through their study on rural Australian communities suffering from prolonged droughts. Residents of these regions reported high levels of solastalgia, manifested in feelings of hopelessness, depression, and a loss of identity connected to the land. Similarly, [10] ([10]) documented solastalgia experiences in Canadian Arctic Inuit communities, whose traditional lifestyles based on hunting and fishing are threatened by the rapid decrease in sea ice. These studies demonstrate that solastalgia is not merely an emotional response to environmental changes but also has profound sociocultural and economic implications.

### Eco-Anxiety, Solastalgia: Lexical Labels for Climate Change-Driven Emotions

According to [16] ([16]), metaphors are central elements of our cognitive structures, functioning as linguistic–conceptual frameworks influencing how we experience, interpret, and act in the world. Beyond being rhetorical tools, these metaphors are deeply integrated into human cognition, organizing our perceptions, emotions, and decisions. The conceptual metaphor theory suggests that much of our abstract thinking is structured through concrete and familiar terms, creating connections that make complex phenomena comprehensible ([16]). These linguistic frameworks not only determine how we understand concepts such as justice, progress, or sustainability but also constrain our response to global problems like the climate crisis. In this context, metaphors become key mediators between language, thought, and action, providing an interpretative structure from which we perceive climate change and articulate emotional and practical strategies to face its challenges.

The cognitive processing of these metaphors can be understood through the use of new lexical elements, namely, a new vocabulary. The management of new vocabulary is associated with representing new meanings influenced by cultural factors shaped through linguistic structures. Understanding messages that include new vocabulary is directly related to the speaker’s exposure to the lexical–cultural process from which these new words emerge. Terms like “eco-anxiety” and “solastalgia” do not merely label an emotional phenomenon but also influence our perception and understanding of climate change’s impact on mental health. Both concepts act as lexical labels encapsulating complex and multifaceted experiences ([2]). These labels not only reflect the emotions of their users but also influence how individuals and society at large perceive and respond to climate change.

Despite being relatively recent lexical elements, the terms eco-anxiety and solastalgia have proliferated on the internet. They are situated within specialized fields, such as environmental psychology and ecology. They are used in academic and professional discussions about the psychological impacts of climate change and environmental degradation. Similarly, their professional and academic links make them common in scientific articles, environmental and mental health forums, and groups specializing in ecopsychology. Additionally, their widespread use implies that both terms are gaining recognition among the general public, with increasing frequency of use extending beyond specialized groups. The coverage of environmental topics on social media and digital media mirrors how people mediately address mental health issues. Both terms have been added to complex, high-intersection psychosocial topics related to the environment. For example, the use of terms such as “climate alarmism” or “catastrophism” by climate change deniers reveals efforts to minimize the urgency of the crisis and attempt to discredit environmental concerns as exaggerations. This language use not only communicates a stance but also shapes the public perception of the problem ([21]). Cognition and mental lexicon are thus intertwined, showing how our words and thoughts interact to form an implicit and meaningful conceptualization of climate change and its effects.

This study aims to analyze the interrelations between eco-anxiety and solastalgia and explore their impact on psychological well-being by analyzing a corpus of messages published on social networks, focusing on Spanish-language media to explore the culturally specific ways in which eco-anxiety and solastalgia are conceptualized, experienced, and communicated within Spanish-speaking digital communities. Using content analysis and lexicometry techniques, this study will examine the discursive configurations and usage patterns of both terms to identify prevailing emotions, coping strategies, and associated sociodemographic factors. We hypothesize that climate-related messages will show a high frequency of terms related to emotional distress, yet they will also reveal elements indicating a search for meaning and the building of supportive communities. This study aims to contribute to a better understanding of how digital discourses can reflect emotional experiences related to the climate crisis and facilitate the construction of hopeful and resilient narratives, which also constitute valuable tools for developing adaptive and preventive resources in mental health.

## 2. Materials and Methods

### 2.1. Design and Sample

The methodological design of this study is qualitative, based on the content analysis methodology. This methodology aims to make valid and replicable inferences using data in their respective contexts, ultimately describing and quantifying specific phenomena ([14]). For the analysis of lexical labels, a corpus of 120 items was compiled, each of which was considered a “publication” for the purposes of this study. A publication was defined as a single digital post or an individual online news article written in Spanish, published between October 2023 and March 2024. The data were collected from the following platforms: Instagram, Facebook, Reddit, X (formerly Twitter), and digital news websites including El País, BBC Mundo, La Tercera, El Mostrador, Infobae, and Página/12. These sources were selected based on their regional relevance and frequency of content addressing mental health and environmental issues in Spanish.

The exclusive focus on Spanish-language content allows for an in-depth examination of cultural and linguistic nuances specific to Spanish-speaking communities. This study does not claim to represent the full diversity of Spanish-speaking digital discourse. Our corpus included both editorially curated publications from established news and press outlets (accessed via their websites or verified social media accounts) and public posts authored by activists and citizens. Although the initial selection of publications was conducted manually, predefined inclusion criteria were strictly applied to minimize researcher bias. Only content that explicitly referenced eco-anxiety or solastalgia in the context of emotional or psychological impact was included, ensuring consistency and transparency in the selection process.

Although the total number of analyzed publications (N = 120) may be considered modest for a digital media content analysis, the sampling strategy emphasized conceptual relevance over volume. Only posts that explicitly mentioned eco-anxiety or solastalgia in relation to mental health, emotional experiences, or psychological responses to environmental change were included. This targeted approach enabled a more focused thematic analysis grounded in clearly defined inclusion criteria, prioritizing depth and interpretive richness. This study did not aim to reach theoretical saturation in the traditional qualitative sense. Instead, it analyzed a bounded corpus of digital publications selected based on predefined inclusion criteria. The goal was to identify recurring lexical patterns and thematic uses of eco-anxiety and solastalgia, rather than to exhaustively capture all possible variations of discourse across platforms.

The lexical screening and initial identification of thematic categories were conducted by two researchers. These preliminary categories were then used to classify the full set of 120 publications by a group of four additional researchers. Although no formal inter-rater reliability index was calculated, discrepancies were discussed in group sessions and resolved by consensus, ensuring consistency in the application of coding criteria across the dataset.

### 2.2. Boolean Search Strategy

To retrieve relevant digital content, we implemented a structured Boolean search strategy. This approach was designed to identify Spanish-language publications explicitly referencing eco-anxiety or solastalgia in the context of mental health or emotional discourse. The search was conducted across online media platforms, including news websites and digital magazines, from October 2023 to March 2023. Table 1 summarizes the full search strategy and inclusion/exclusion criteria applied.

Posts or articles were excluded if they mentioned “eco-anxiety” or “solastalgia” in a purely political, economic, or technical context, without reference to emotional, psychological, or experiential dimensions. For example, content analyzing eco-anxiety solely as a rhetorical strategy in political campaigns or economic policy debates—without acknowledging its mental health implications—was not included. This exclusion was based on the study’s objective to explore the emotional resonance and subjective experience of these terms. Other exclusion criteria included posts consisting only of images or videos without text, advertising content, satire, or irrelevant uses of the terms detached from environmental or psychological themes.

This search procedure was initially conducted manually and then by using the software LIWC-22 (Linguistic Inquiry and Word Count, version 2022), which identifies linguistic patterns and frequencies of specific words in large textual datasets. The analysis of the various publications meeting the search criteria generated a series of text segments, which, after analyzing them within their respective semantic fields, led to the emergence of four main categories that encompass the principal representations of climate change. As part of the initial screening phase, we used LIWC software to identify posts with elevated frequencies of affective and anxiety-related terms. This automated step supported the identification of emotionally relevant content for inclusion, but no formal LIWC-based analysis was performed or reported.

These four thematic categories were developed inductively through a qualitative coding process guided by grounded theory principles. The research team began by reviewing a sample of 30 posts to identify recurring emotional and discursive patterns. Two coders independently proposed initial codes, which were then discussed and consolidated into a codebook. The entire dataset was coded based on this shared framework.

### 2.3. Analytical Framework

An inductive qualitative content analysis was employed. The coding process followed grounded theory principles, allowing thematic categories to emerge from the data. Two researchers independently coded an initial subsample and refined the codebook collaboratively. The final categories were: (1) environmental awareness activism, (2) testimony and emotional expression, (3) scientific and technical discourse, and (4) political–ideological positioning. Each publication was assigned to only one category to preserve mutual exclusivity.

After initial lexical screening, researchers engaged in the open coding of the selected texts, identifying recurring semantic patterns and emergent themes. This process led to the creation of five final categories, which were refined through iterative discussion and application by multiple coders. The approach prioritized thematic saturation and interpretative depth rather than theory-driven coding, allowing the data to inform category construction.

The relative distribution of categories is presented descriptively to illustrate the prevalence of themes within this specific corpus. These proportions are not intended to support statistical comparisons or significance testing.

This study offers an initial exploration of how eco-anxiety and solastalgia are represented in Spanish-language digital discourse. However, it does not include triangulation with additional data sources or methods. Future studies could compare findings across multiple social media platforms, time periods, or analytic techniques (e.g., sentiment analysis, survey data, or correlation with climate events) to provide a more comprehensive understanding of this phenomenon.

### 2.4. Data Management

All retrieved content was stored in a database. Coded data were organized in spreadsheets, and coding consistency was reviewed collaboratively by the research team. Regular meetings ensured category agreement and thematic coherence.

### 2.5. Ethical Considerations

This study analyzed publicly available content that did not include personal or sensitive data. In accordance with institutional guidelines and national regulations, no ethics committee approval was required. Nonetheless, all sources were cited appropriately to respect intellectual property.

## 3. Results

In this study, we have delved into participant responses related to climate change. The emerging categories provide not only data but also a window into the attitudes and emotions surrounding this crucial issue. We will present findings related to the lexical labels on eco-anxiety and solastalgia, classified into the four aforementioned categories, in descending order of their recurrence in the corpus. Additionally, we will provide evidence about the roles with which users of digital platforms identify themselves.

The results of the analysis of lexical labels related to climate change can be visualized through the following lexical label charts on eco-anxiety and solastalgia, which illustrate the distribution of word categories used in the publications. The lexical labels are grouped into the areas of “catastrophic thinking or fatalism,” “loss of places of personal significance,” “environmental awareness activism,” and “coping strategies.” The most prominent category is that of environmental activism, closely followed by catastrophic thinking. Alongside this, we include a word cloud representing the most frequent words associated with eco-anxiety and solastalgia within the corpus of analyzed publications (Figure 1). The distribution of lexical labels by thematic category and frequency is summarized in Table 2.

These visual tools effectively convey the dominant themes and emotional tones prevalent in the discussions surrounding climate change’s psychological impacts. The word cloud specifically highlights the recurrent vocabulary and concepts, providing insight into the emotional and cognitive frameworks experienced by individuals engaged with these concerns. It serves as a powerful representation of the collective consciousness regarding eco-anxiety and solastalgia, emphasizing the nuanced and complex interplay of fear, activism, and coping in the face of a changing climate.

### 3.1. Categories of Eco-Anxiety and Solastalgia

Based on the inductive analysis, we identified four thematic categories: (1) environmental awareness activism, (2) testimony and emotional expression, (3) scientific and technical discourse, and (4) political–ideological positioning. Each publication was assigned to one of these mutually exclusive categories, which are described and illustrated below. These categories capture the range of emotional responses and discursive patterns observed in Spanish-language content related to climate change. Their interrelationships, as well as their role in shaping public discourse and emotional engagement, are summarized in Figure 2.

Catastrophic Thinking or Fatalism: This category refers to a tendency to anticipate and worry about the worst possible outcomes of an event or situation, especially regarding climate change. Individuals with this type of thinking often feel that the negative impacts of climate change are inevitable and devastating, and that their personal efforts cannot make a significant difference.

Loss of Places of Personal Significance: This focuses on the emotional distress resulting from the degradation or destruction of places that hold deep personal meaning for an individual. These places may include one’s home, childhood sites, favorite natural areas, or any environment with significant emotional or cultural value.

Environmental Awareness Activism: This involves active participation in efforts to increase public awareness about environmental issues and promote actions to combat climate change. This type of eco-anxiety motivates individuals to engage in advocacy and environmental education activities.

Coping Strategies: Coping strategies are methods and techniques that individuals use to manage and reduce anxiety and stress caused by climate change. These strategies can be both personal and collective, and they aim to promote emotional well-being and resilience.

Catastrophic Thinking or Fatalism and Loss of Places of Personal Significance: By utilizing “AND,” it was possible to find publications addressing both eco-anxiety and solastalgia simultaneously. These terms interrelate in the matrix since catastrophic thinking is often linked with the perception of loss of significant places. This Boolean code allows the identification of studies addressing this interplay. For example, an individual anticipating the worst outcomes of climate change may experience intense eco-anxiety, especially if their places of personal significance are at risk or have already been affected.

Concerning coping strategies, the use of “AND” enabled the identification of how individuals manage eco-anxiety related to solastalgia. In other words, it highlighted how people develop coping strategies to mitigate the emotional impact of losing places of personal significance due to climate change.

Catastrophic Thinking or Fatalism and Environmental Awareness Activism: Using “OR” in this context extended the search to include studies discussing either concept or both. This was useful to capture a broader range of publications more focused on eco-anxiety (and how it leads to either catastrophic thinking or activism as a response) or on solastalgia (and how the loss of significant places influences mental health).

Regarding coping strategies, the publications found through “OR” offered perspectives on different forms of coping strategies, whether they apply specifically to eco-anxiety, solastalgia, or both.

Each post or article was assigned to a single, mutually exclusive category based on the dominant emotional or thematic focus identified during coding. This approach enabled a more structured comparison of the prevalence of each discourse type across the corpus.

The five discursive strategies—emotionalization, dramatization, activation, polarization, and legitimation—were adapted from the analytical framework outlined in the ECODES Observatory report. These categories served as a second-level coding scheme applied to the previously classified content, allowing for a more nuanced understanding of the rhetorical mechanisms present in the texts.

### 3.2. Roles of Users Posting About Eco-Anxiety and Solastalgia

Citizens: Ordinary individuals without specific professional or strategic affiliations who express personal concerns or insights related to eco-anxiety or solastalgia. They often share emotional reactions or personal stories.Politicians and Government Officials: Those involved in policymaking or governance often share information or policies related to climate change and its implications on mental health.Young Activists: Individuals, often from younger generations, actively participating in environmental movements, advocacy, and raising awareness regarding eco-anxiety and solastalgia.Entrepreneurs: Business figures who focus on sustainable practices, the economic implications of climate change, or corporate responsibility in mitigating associated mental health impacts.Scientists: Researchers and academics who provide data-driven insights, studies, and expert opinions on the relationship between climate change, eco-anxiety, and solastalgia.

Each role contributes a unique perspective to the discourse surrounding eco-anxiety and solastalgia, reflecting diverse concerns, solutions, and priorities within the broader context of the climate crisis. These roles illustrate the diversity of perspectives and discourses in digital media regarding eco-anxiety and solastalgia (Figure 3).

### 3.3. Environmental Awareness Activism

This category encompasses the majority of the publications, accounting for 41% of the responses. The highlighted characteristics include environmental concerns and the anxiety they provoke, which serve as the motivation for their activism. These users actively work to inform and raise awareness among others about the importance of environmental protection and climate change mitigation:

“We cannot remain indifferent while the planet dies. Every small action counts—our voice, our protest, our daily choices.”

Within these publications, we find a diversity of actors, including various groups, young activists, ordinary citizens, scientists, intellectuals, and indigenous communities ([12]). These users share information about a wide range of activities, from individual awareness about reducing ecological footprints to educational projects aimed at the general population. Additionally, they engage in battles against large corporations, organize protests and campaigns, and participate in political and legislative management to mitigate climate change.

Predominantly, these users are motivated by a profound awareness of environmental degradation and the sense of responsibility that befits us as human beings. The focus of most of these publications is on transforming life models, development, and energy consumption, promoting a change in thinking overall. These activists believe it is critical to transform daily practices and adopt more sustainable lifestyles to curb the impact of climate change.

The publications associated with anxiety reflect individual awareness and the promotion of practices that reduce the ecological footprint, such as recycling, using renewable energies, and reducing plastic consumption. They also emphasize the promotion of educational projects, implementing programs to educate the population about the importance of the environment and how each person can contribute to its conservation.

Moreover, we find publications related to the fight against large corporations, denouncing predatory practices and advocating for laws regulating the impact of such enterprises. Lastly, there are calls to participate in protests and campaigns to capture public attention and pressure governments and companies to take concrete action against climate change. These users are committed to creating a more sustainable and equitable future where individual and collective actions contribute to the preservation of the planet for future generations.

### 3.4. Catastrophic Thinking or Fatalism

The lexical labels related to climate change falling under this category account for 29% of user responses. We detect a greater participation from young activists and youths in general, whose publications primarily testify to their personal experiences, focusing on extreme concern about the negative consequences associated with climate change. Moreover, these publications recurrently relate to the idea of the inevitability of an eventual climate collapse. A tendency to refer to the worst possible scenarios associated with climate change was identified. This is often accompanied by explicit expressions of powerlessness in the face of climatic events, along with negative emotions such as extreme anxiety, fear, hopelessness, and sadness due to the perception of a bleak future against the backdrop of a global climatic deterioration perceived as irreversible:

“There’s no future. Everything’s collapsing and no one listens. I’m overwhelmed by fear and guilt every time I read the news.”

It is important to note that 29% constitutes a significant portion of digital expressions on this subject, and the relationship between fatalism and climate inaction, especially among the youth, is concerning. Undoubtedly, this category appears to be the most detrimental to mental health due to the intensity of the negative emotions detected. Concepts such as “deterioration,” “destruction,” “distress,” “frustration,” and “guilt,” among others, reflect a grim and immobilizing outlook.

### 3.5. Coping Strategy

Representing 25% of the responses, this category encompasses coping strategies mentioned primarily by citizens, affected communities, and mental health specialists seeking ways to deal with the emotional impact of climate change ([15]). Proposals often include adopting self-care practices such as talking with friends or participating in support groups. These coping strategies are crucial for managing anxiety and stress stemming from growing environmental concerns.

We also find publications promoting social support through networks of friends, family, and communities with shared environmental concerns. Belonging to these groups provides a safe space to share experiences and strategies, which can alleviate feelings of isolation and increase collective resilience. Connecting with others who understand and share these concerns can strengthen the determination to face environmental challenges together.

These publications often assert that engaging in concrete actions can provide a sense of contribution and problem mitigation. Sustainable practices such as recycling, reducing energy consumption, and adopting eco-friendly habits are promoted as tangible ways to address climate change. These actions are highlighted not only for their positive environmental impact but also as strategies that enhance individuals’ sense of agency by making them feel proactive in addressing the crisis. This sense of agency seems to counteract feelings of powerlessness and despair, providing clear purpose and direction.

Overall, these coping strategies, which include personal approaches, social support, and concrete actions, are proposed as essential for managing the emotional impact of climate change.

### 3.6. Loss of Places of Personal Importance

This category recorded only 6% of the responses. Primarily, it revealed explicit expressions of the intense emotional pain caused by the loss of places that form part of the user’s identity and memory ([3]). The publications reflect deep feelings of displacement and lack of belonging, as well as palpable sadness over the damage or disappearance of significant places.

The emotional pain associated with losing these places is linked to the rupture of personal and community ties. Users express how these spaces, once the backdrop for important and memorable moments, have deteriorated or vanished due to climate change:

“My grandmother’s house was surrounded by trees. They’re all gone now. That place was my childhood, and it feels like I lost it too.”

This loss impacts not only the environment but also deeply affects the emotional and social fabric of individuals.

Manifestations of displacement reflect a sense of disconnection from the environment that once provided a sense of belonging ([19]). The destruction of these places evokes a sense of emptiness and nostalgia, heightening emotional distress and feelings of helplessness. Additionally, the sadness over the damage or disappearance of these sites underscores the importance of the bonds between individuals and their environments, highlighting how climate change alters not only the physical but also the emotional and cultural landscapes of communities.

## 4. Discussion

Eco-anxiety, although not considered a mental illness per se, is a phenomenon with the potential to negatively impact the mental health of those who experience it. It can also result in psychological effects of varying severity in some individuals, particularly in children and young people who may be particularly vulnerable due to their high exposure to digital media, which are sources that generate eco-anxiety and solastalgia. Social media plays a dual role in this context; on one hand, it fosters the dissemination of information and promotes collective awareness and action. On the other hand, it can amplify feelings of eco-anxiety and ecological grief, especially when users are constantly exposed to negative news or impactful images of environmental degradation. This underscores the importance of understanding the specific dynamics through which social media influences mental health related to climate change and other environmental issues.

The results of the study on eco-anxiety and solastalgia emphasize the significant influence of lexical labels in the perception and management of emotions associated with climate change. These labels not only describe complex emotional phenomena but also guide how individuals and society understand and respond to environmental crises. This analysis prompts us to reflect on the importance of addressing both the emotional and rational aspects of climate change. As a society, it is crucial to develop a deeper environmental awareness that encourages effective climate action, thus ensuring a sustainable future for the coming generations.

It is important to recognize that fatalism can be an obstacle to climate action, especially among younger populations who express high levels of climate-related distress ([8]). Although the severity of the climate crisis is undeniable, it is essential to identify pathways to transform anxiety and despair into meaningful engagement and constructive responses ([22]). The challenge lies not only in recognizing the psychological impact of environmental degradation but also in promoting coping strategies that enhance agency and resilience ([17]).

Awareness of this urgency should not paralyze us but rather propel us to take concrete steps to mitigate the effects of climate change. Individuals adopting these practices are demonstrating notable resilience and a profound commitment to creating a more sustainable and equitable future. By integrating these strategies into their lives, they are not only caring for their emotional well-being but also actively contributing to the preservation of the planet for future generations.

On the other hand, as climate-related problems worsen, the number of people experiencing eco-anxiety will also increase, given that a larger portion of the population will experience firsthand the effects of climate change, such as extreme weather events, natural disasters (e.g., floods, droughts, and hurricanes), deforestation, and large-scale fires causing prolonged environmental transformations, thereby broadening the discussion of the topic on social media.

Among the groups most affected by eco-anxiety are young people, indigenous communities, and those with a close connection to nature. The literature shows that these groups may experience emotional, psychological, and spiritual impacts due to climate change. However, there is a noted lack of representation of certain regions of the world and indigenous populations in existing research.

It is imperative to extend research to young populations and conduct more quantitative studies that include participants from regions outside Northern Europe, North America, and Australia, considering pertinent exceptions. In survey studies, it is crucial to analyze in detail other specific subgroups to identify those with particular vulnerabilities.

### 4.1. Comparisons with Other Research

Research on the representation of climate change in the media and its emotional impact reveals complex patterns that connect communication, collective emotions, and social responses. [11] ([11]) compared climate change news coverage in two Chilean newspapers with different ideological orientations. In the liberal medium, news was more frequent, extensive, and with a greater diversity of perspectives, suggesting that editorial ideologies significantly influence the depth and plurality of the climate debate. This finding highlights how media interpretive frameworks affect public perception and can contribute to either a richer or more limited understanding of environmental issues.

The predominance of environmental awareness activism in our corpus (41%) suggests a strong discursive orientation toward mobilization and climate engagement. This aligns with recent literature highlighting eco-anxiety not only as a psychological response but also as a driver of collective environmental action and eco-citizenship ([24]). Rather than paralyzing individuals, eco-anxiety can become a motivating force that channels emotional distress into civic participation, particularly among younger demographics ([22]).

On the other hand, [9] ([9]) highlight key differences between scientific publications and mass media in Chile. While the former often focuses on the ecological consequences of global warming, mass media present a broader range of approaches, including political, economic, and social narratives. This contrast not only evidences differences in target audiences and their expectations but also how messages can influence climatic emotions, from concern to activism.

In a more specific analysis of the actors behind the media agenda, [13] ([13]) detected that, in Chile, government entities are the main definers of climate topics in the media, in contrast to the lesser prominence of citizen organizations. This centralization of discourse could limit the representation of diverse voices, particularly those that might foster alternative narratives or more local and community-focused approaches to climate action.

Internationally, [25] ([25]) provide a psychological perspective by exploring how eco-anxiety influences pro-environmental behavior among Chinese university students. Their research suggests that, although eco-anxiety can motivate environmental actions, individuals showing resilience in the face of climate news tend to experience less eco-anxiety and, surprisingly, also less pro-environmental engagement. This raises questions about the relationship between perceptions of climate risk, associated emotions, and resultant behaviors. While resilience appears to mitigate the emotional burden, it might also dampen the sense of urgency needed to foster meaningful climate action.

### 4.2. Expansion of Proposed Ideas

These studies highlight the importance of analyzing cultural, political, and emotional factors mediating the media representation of climate change and individual and collective responses. Findings by [11] ([11]) and [9] ([9]) underscore how discursive frameworks vary by medium and region, implying that climate communication strategies must be adapted to local contexts to maximize their effectiveness. For example, more inclusive and plural coverage could foster a greater sense of agency among readers, promoting climate action through a narrative combining scientific data with personal and community stories.

The observation by [13] ([13]) about governmental predominance in defining climate topics invites reconsideration of how media agendas are structured. Involving citizen organizations and local communities would not only diversify perspectives but could also increase the relevance and emotional impact of climate stories.

Regarding the study by [25] ([25]), it presents a key challenge: how can we balance the emotional urgency needed to mobilize climate action with strategies promoting resilience and mental well-being? A possible solution is to integrate educational approaches that highlight not only the risks but also opportunities for collective action, emphasizing examples of success and positive adaptations to climate change.

In summary, the findings presented indicate that media representation of climate change and associated emotions have profound implications for mental health, risk perception, and climate action. Integrating these dimensions into an interdisciplinary approach can contribute to designing more effective and contextually relevant interventions to tackle the challenges of the climate crisis.

As in all qualitative research, the interpretation of findings is shaped by the perspectives of the research team. The interdisciplinary background of the team—including psychology, linguistics, and psychiatry—enabled diverse viewpoints and ongoing discussion of potential interpretive biases. Regular team meetings and joint review of the coding framework supported reflexivity and analytical rigor.

### 4.3. Limitations and Future Directions

This study has some limitations that should be acknowledged. First, the analysis was restricted to Spanish-language digital media, which may limit the generalizability of the findings to broader linguistic and cultural contexts. Additionally, although content was systematically categorized, the absence of intercoder reliability testing may introduce some subjectivity in the coding process. The focus on news and digital press sources, while valuable, excludes user-generated content such as social media, which could provide further insight into public sentiment. Future research should consider comparative analyses across different languages and media ecosystems, explore the evolution of eco-anxiety discourse over time, and incorporate metaphor analysis to better understand how language mediates environmental emotions.

Another limitation of this study is a lack of systematic geographic breakdown of the digital content. Although the analysis focused on Spanish-language publications, the transnational nature of many platforms made it difficult to reliably assign content to specific countries or regions. Future research could incorporate regional filters to explore how eco-anxiety and solastalgia are framed in different Spanish-speaking countries. Absence of participant validation (e.g., member checking), which was not feasible due to the anonymous and publicly sourced nature of the digital content, is another limitation. As such, the interpretations presented rely solely on the researchers’ thematic analysis and are not corroborated by the original authors of the posts.

To further advance this field, future research would benefit from qualitative designs capable of capturing the lived experiences, metaphors, and contextual meanings embedded in eco-anxiety discourse. In particular, mixed-methods studies could combine large-scale lexical analyses with in-depth interviews or ethnographic approaches, providing a more comprehensive understanding of how individuals and communities emotionally and cognitively process environmental threats.

## 5. Conclusions

In a global context marked by the rise of environmental issues, it is crucial to understand how associated emotions impact mental health. Public reactions to climate change are evolving, highlighting the crucial role of social networks in raising awareness. These platforms not only increase exposure to climate impacts through information and testimonials but also foster dialogue on environmental concerns, intensifying individual and collective emotions.

International research shows widespread concern over climate change, underscoring the need to better understand these emotions and their psychological effects. It is essential to explore how these emotions can catalyze action or contribute to feelings of helplessness. In light of this, we advocate for a comprehensive approach that analyzes key emotions such as concern, anxiety, and grief, especially in digital contexts. Understanding these dynamics is fundamental to designing interventions that promote constructive management and psychological resilience in the face of the ecological crisis, effectively communicating these findings to broader audiences beyond the scientific domain.

## Figures and Tables

**Figure 1 behavsci-15-01102-f001:**
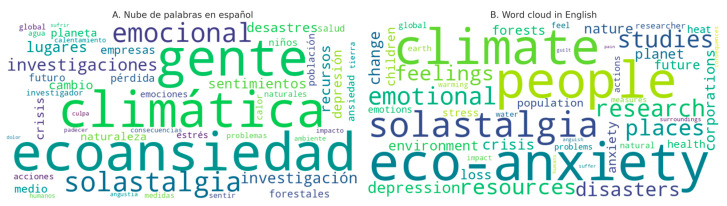
Word cloud showing the most frequent lexical labels related to eco-anxiety and solastalgia in Spanish-language digital media posts. **Left**: Spanish version; **right**: English version.

**Figure 2 behavsci-15-01102-f002:**
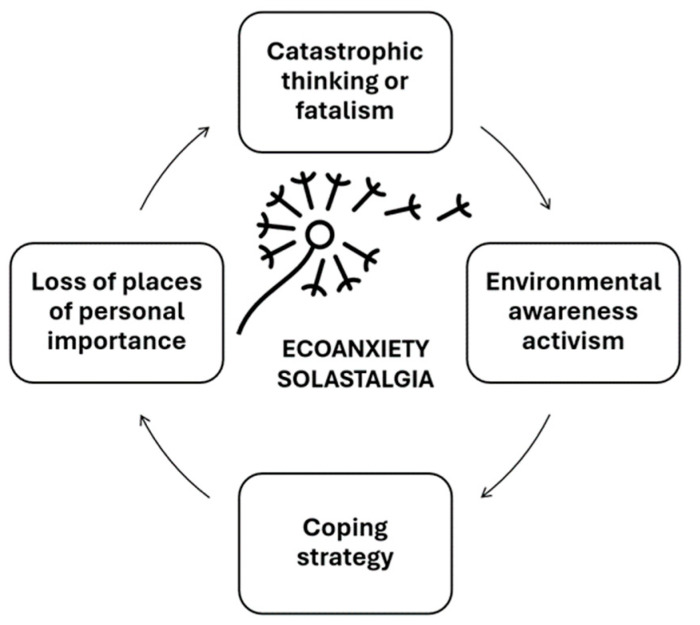
Categories of eco-anxiety and solastalgia.

**Figure 3 behavsci-15-01102-f003:**
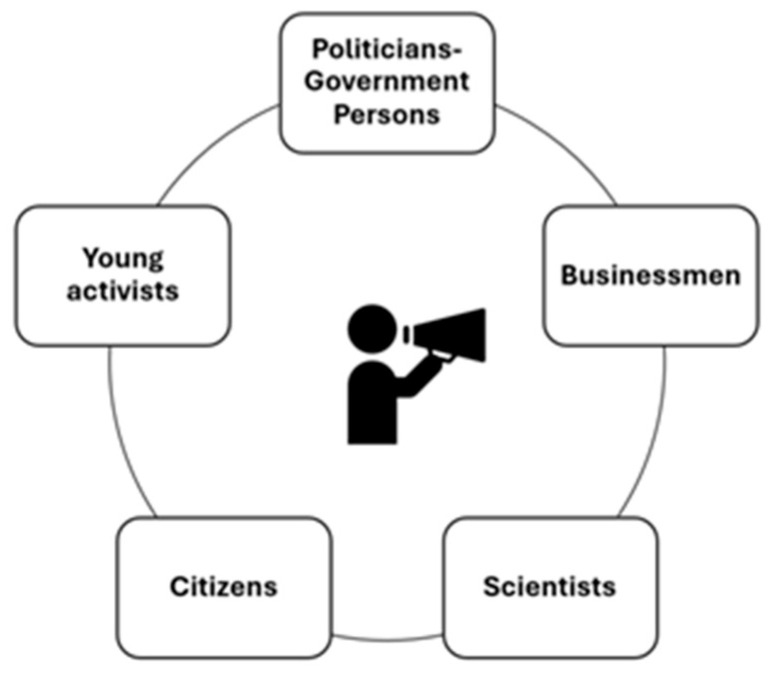
Roles of users posting about eco-anxiety and solastalgia.

**Table 1 behavsci-15-01102-t001:** Boolean search strategy and inclusion/exclusion criteria.

Component	Details
Search terms	(“eco-anxiety” OR “solastalgia”) AND (“mental health” OR “emotion” OR “feelings”)
Language	Spanish
Date range	October 2023–March 2024
Sources	Online newspapers, magazines, digital news media. Social media (e.g., Twitter, Facebook, Instagram)
Exclusion criteria	Publications focused exclusively on economic or political commentary unrelated to mental health
Inclusion criteria	Publications that explicitly mention eco-anxiety or solastalgia in a mental health or emotional context

**Table 2 behavsci-15-01102-t002:** Distribution of lexical labels related to eco-anxiety and solastalgia by thematic category and frequency in Spanish-language digital media posts.

Eco-Anxiety		
Category	Number of Entries	Percentage (CI)
Catastrophic thinking or fatalism	20	29% (18.7–41.2)
Loss of places of personal importance	4	6% (1.6–14.2)
Environmental awareness activism	28	41% (28.9–53.1)
Coping strategy	17	25% (15.1–36.5)
Total	69	100%
**Solastalgia**		
Catastrophic thinking or fatalism	2	4% (0.4–12.3)
Loss of places of personal importance	22	39% (26.5–53.2)
Environmental awareness activism	17	30% (18.8–44.1)
Coping strategy	15	27% (15.8–40.3)
Total	56	100%

Total sample includes 120 digital publications. Multiple lexical labels may appear within a single publication.

## Data Availability

The data presented in this study are available upon request from the corresponding authors due to technical restrictions.

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
