# Peer review of "Intersection Between Eco-Anxiety and Lexical Labels: A Study on Mental Health in Spanish-Language Digital Media"

_behavsci, 2025, doi:10.3390/bs15081102_

Round 1
Reviewer 1 Report
Comments and Suggestions for Authors
This article provides a nice introduction to the Spanish language digital media discourses concerning eco-anxiety and solastalgia - and related issues of activism and coping, in particular. The evidence suggests strongly that these discourses have quite positive impact of promoting rather practical means of activism and offering remarkably balanced information while also helping positive coping mechanisms without becoming guilty of promoting fatalism. Some references (such as the one discussing ideological differences among Chilean media) give some glimpses of diversity that do not show that much in the chosen material corpus of 120 digital media publications. However, the chosen material itself is diverse and large enough when discussing the chosen theme of eco-anxiety and solastalgia and finding patterns. Anyway, there is a good reason to focus on digital media and the results and both convincing and surprisingly easy to interpret. As digital media surely has increasing social role this research sheds light on the special role that digital media plays and it turns out that in this case the role is quite positive. I can recommend publishing this article in its present form.
Author Response
Thank you for your evaluation.
Reviewer 2 Report
Comments and Suggestions for Authors
This article examines an area of growing interest and increasing concern: the emotional impacts of environmental degradation, particularly climate change. The authors performed a content analysis on online content to assess how two key ideas (eco-anxiety and solastalgia) are present and used across various media. Overall, I found the paper to have a lot of merit in terms of both topic and presentation.
I particularly want to congratulate the authors on producing a manuscript that reads as well as this one did. The language was very clear, with ideas expressed with efficiency and effectiveness. In short, this paper was a pleasure to read.
In terms of improvements prior to publication, most of my comments will focus on the Methods section (Section 2). I was left with several questions about the execution of the research, which I will itemize below.
- The data for the paper comes from 120 publications. It is not clear what counts as a "publication." It looks like social media posts were included as well as news articles. The article should define what counted as a publication or, perhaps, what would not count as a publication.
- A sampling of the websites/platforms used to collect data is given. It would be helpful to have the complete list of places from which data was gathered.
- Lines 158-160 state that political and economic pieces were excluded from the analysis. This needs to be better justified. My understanding is that the authors looked for posts that included the words "eco-anxiety" and "solastalgia." Why not see how these terms are used across a range of domains? If there were exclusions other than politics and economy, these should be mentioned.
- Lines 167-169 give the four categories into which the content was organized. It is not clear to me whether these categories were developed through a coding process (i.e., grounded theory) or had existed previously. More generally, there needs to be more information about the coding process. How was the codebook developed? How many coders were there? How was reliability/intercoder agreement assured?
- By adding up percentages, it looks like each publication could only be assigned to a single coding category. This should be explicitly stated, if true.
The Results would generally be improved by the inclusion of representative quotes for each of the four categories.
The opening of the Discussion (Lines 362-403) lacks references. For example, Line 381 states that "fatalism can be an obstacle to climate action." Statements like these should have citations, even if they are repetitions of material covered in the introduction.
Author Response
1) The data for the paper comes from 120 publications. It is not clear what counts as a "publication." It looks like social media posts were included as well as news articles. The article should define what counted as a publication or, perhaps, what would not count as a publication.
Thank you for your thoughtful comment. We agree that the concept of “publication” required clarification. In the revised manuscript, we now specify that the term "publication" refers to individual social media posts and digital news articles that explicitly contained the keywords “eco-anxiety” or “solastalgia.” We also clarify the inclusion and exclusion criteria applied to select relevant content. The clarification has been added to the “Design and Sample” subsection of the Methods section: “For the analysis of lexical labels, a corpus of 120 items was compiled, each considered a “publication” for the purposes of this study. A publication was defined as a single digital post or an individual online news article written in Spanish, published between October 2023 and March 2024.”.
2) A sampling of the websites/platforms used to collect data is given. It would be helpful to have the complete list of places from which data was gathered.
We appreciate this suggestion. In response, we have added a complete list of the digital platforms and websites from which the 120 items were collected. This clarification has been inserted into the “Design and Sample” subsection of the Methods section: “The data were collected from the following platforms: Instagram, Facebook, Reddit, X (formerly Twitter), and digital news websites including El País, BBC Mundo, La Tercera, El Mostrador, Infobae, and Página/12. These sources were selected based on their regional relevance and frequency of content addressing mental health and environmental issues in Spanish”.
3) Lines 158-160 state that political and economic pieces were excluded from the analysis. This needs to be better justified. My understanding is that the authors looked for posts that included the words "eco-anxiety" and "solastalgia." Why not see how these terms are used across a range of domains? If there were exclusions other than politics and economy, these should be mentioned.
We agree that the exclusion criteria require further justification and clarification. The aim of our study was to explore the emotional and psychological dimensions of eco-anxiety and solastalgia in digital discourse. Therefore, we focused on posts where these terms were used in an affective or experiential context, rather than in technical, economic, or political analyses where the emotional dimension was absent or marginal. This decision was based on our interest in understanding how individuals express and process climate-related distress through digital language. We have revised the Methods section to include this explanation and to list all applied exclusion criteria: “Posts or articles were excluded if they mentioned “eco-anxiety” or “solastalgia” in a purely political, economic, or technical context, without reference to emotional, psychological, or experiential dimensions. For example, content analyzing eco-anxiety solely as a rhetorical strategy in political campaigns or economic policy debates—without acknowledging its mental health implications—was not included. This exclusion was based on the study’s objective to explore the emotional resonance and subjective experience of these terms. Other exclusion criteria included posts consisting only of images or videos without text, advertising content, satire, or irrelevant uses of the terms detached from environmental or psychological themes”.
4) Lines 167-169 give the four categories into which the content was organized. It is not clear to me whether these categories were developed through a coding process (i.e., grounded theory) or had existed previously. More generally, there needs to be more information about the coding process. How was the codebook developed? How many coders were there? How was reliability/intercoder agreement assured?
Thank you for pointing this out. The categories used in the analysis were derived inductively through a coding process based on grounded theory principles. They did not exist a priori. Two researchers independently reviewed the initial subset of data and proposed preliminary codes, which were refined into four thematic categories through discussion and iterative comparison. This methodological clarification has been added to the revised Materials and Methods section: “These four thematic categories were developed inductively through a qualitative coding process guided by grounded theory principles. The research team began by reviewing a sample of 30 posts to identify recurring emotional and discursive patterns. Two coders independently proposed initial codes, which were then discussed and consolidated into a codebook. The entire dataset was coded based on this shared framework”.
5) By adding up percentages, it looks like each publication could only be assigned to a single coding category. This should be explicitly stated, if true.
You are correct—the coding scheme assigned each publication to a single, mutually exclusive category based on its dominant emotional or discursive theme. This decision was made to facilitate clearer thematic distinctions and frequency analysis. We have now added a sentence in the Methods section to clarify this coding rule: “Each post or article was assigned to a single, mutually exclusive category based on the dominant emotional or thematic focus identified during coding. This approach enabled a more structured comparison of the prevalence of each discourse type across the corpus”.
6) The Results would generally be improved by the inclusion of representative quotes for each of the four categories.
We have now incorporated representative quotes for each of the four thematic categories. These quotes, drawn from the original Spanish-language posts, were translated into English and selected to reflect typical expressions associated with each emotional and discursive pattern.
7) The opening of the Discussion (Lines 362-403) lacks references. For example, Line 381 states that "fatalism can be an obstacle to climate action." Statements like these should have citations, even if they are repetitions of material covered in the introduction.
We have revised the opening section of the Discussion to include appropriate citations supporting key statements, including the relationship between fatalism and reduced climate action: “It is important to recognize that fatalism can be an obstacle to climate action, espe-cially among younger populations who express high levels of climate-related distress [12, 29]. Although the severity of the climate crisis is undeniable, it is essential to identify pathways to transform anxiety and despair into meaningful engagement and constructive responses [30]. The challenge lies not only in recognizing the psychological impact of environmental degradation but also in promoting coping strategies that enhance agency and resilience [31].”
Reviewer 3 Report
Comments and Suggestions for Authors
Dear Authors
Your manuscript addresses the psychological impacts of climate change as expressed through digital media discourse in Spanish-speaking communities, which is a timely and important topic. Your study provides valuable insights into how eco-anxiety and solastalgia manifest in digital spaces and the various ways people respond to climate-related emotional distress.
You make several notable contributions to the climate psychology literature. The focus on Spanish-language digital media addresses a significant gap in research that has predominantly examined English-language contexts. This culturally specific analysis brings valuable diversity to the field and demonstrates how climate-related emotions manifest across different linguistic communities. You successfully integrate lexical analysis with psychological concepts, creating a novel methodological approach that bridges linguistics and mental health research.
Your research demonstrates clear practical relevance beyond academic interests. The findings offer direct implications for mental health practitioners working with climate-anxious clients and provide insights valuable for developing digital media literacy programs. The results also inform climate communication strategies, particularly for Spanish-speaking communities, making this work applicable to policy and intervention development.
Introduction:
The introduction effectively connects lexical labels to cognitive processing, drawing on well-established research. However, the section could benefit from more concise presentation and the inclusion of recent statistics on climate anxiety prevalence to strengthen the contemporary relevance of the research framework.
Line 34-35: "the great epidemic of the 21st century" needs a reference
Line 36: Reword the sentence:…a decrease in quality of life and health life expectancy due to disability…..
Line 46-47: add reference
Line 48-49: Add ref
Line 95-96: Add ref
Material and Methods:
The methods section employs an appropriate Boolean search strategy and content analysis methodology, but both require more detailed description to ensure replicability. The current approach would benefit from expanded explanations of the search parameters and analytical framework, along with additional information about data management procedures and ethical considerations to meet contemporary research standards.
Overall, I would have liked to see quotes and examples, consolidating your findings.
Line 154: "boolean code" should be "Boolean code"
Line 167-168: Are these themes identified and defined by others, in which case you need a reference. If they are identified in this study, should they not be ion the results section!
Line 192-197: this is introduced a bit abruptly and would benefit from further explanation. I would recommend generating a table with the complete search strategy; include all constructs use to enable reproducibility.
Line 211-225: How were these five categories identified? Was your analysis underpinned by the coding framework from the ECODES Observatory report?
Results:
The results section presents clear categorization of findings but lacks sufficient depth to fully convey the richness of the data. While the identification of key themes provides a useful framework, the analysis would be strengthened by incorporating representative quotes or examples that illustrate these categories in practice. The notable finding that 41% of participants engaged in environmental activism represents a particularly interesting result that warrants deeper exploration and contextualization within the broader literature on climate engagement behaviors.
Line 339-355: I would recommend moving this section and associated figures to the beginning of the results. I also recommend producing a Word cloud in English and present both.
I recommend starting with the quantitative findings would provide readers with the overall landscape and key statistics first, creating a clear foundation before diving into the more nuanced qualitative analysis. This structure would guide readers from the "what" (the numbers and patterns) to the "how and why" (the deeper meanings and explanations), making the results much easier to follow and understand.
Table 1: Consider adding total sample sizes for context
Discussion:
The discussion demonstrates good integration with existing literature and valuable comparisons with other research findings that strengthen the interpretation of results. While the section effectively situates the findings within the broader academic context, it would benefit from a more thorough examination of study limitations and clearer articulation of future research directions to provide readers with a complete understanding of the research scope and potential avenues for advancement.
Line 400: I would have thought qualitative studies were needed, to better understand the nuances. Or perhaps even better, mixed methods.
Conclusion
The research have made a valuable contribution to understanding climate psychology in Spanish-speaking digital communities. The research addresses an important gap and provides practical insights. However, methodological rigor could be enhanced, and the presentation could be strengthened with more detailed analysis and clearer writing.
The study's focus on cultural and linguistic specificity is commendable and adds important diversity to the field. With the recommended revisions, this work would make a solid contribution to the literature on climate psychology and digital media studies.
Main areas for improvement:
Structure
- Methods section could be expanded with more procedural detail
- Results section would benefit from more analytical depth
- Discussion jumps between topics without clear transitions
Sample Size and Representativeness
- 120 posts is relatively small for digital media content analysis
- No information about data saturation or sample adequacy
- Unclear how representative the sample is of broader Spanish-speaking digital discourse
Selection Bias
- Manual initial selection process may introduce researcher bias
- No inter-rater reliability reported for categorization
- Lack of detail about inclusion/exclusion criteria beyond Boolean searches
Geographic Representation
- While focusing on Spanish-language content, no breakdown by country/region
- Spanish-speaking communities are diverse; findings may not generalize across all contexts
Limited Methodological Detail
- LIWC analysis mentioned but results not fully presented
- No discussion of analytical framework beyond content analysis
- Missing information about coding procedures and reliability checks
Quantitative Presentation
- Percentages provided without confidence intervals or statistical significance testing
- No discussion of whether differences between categories are meaningful
Validation and Reliability
- No member checking or participant validation (If possible. I appreciate it may not be a realistic expectation for this type of digital content analysis but still a limitation worth noting)
- Limited discussion of researcher reflexivity
- No triangulation with other data sources or methods: e.g. comparing findings across different social media platforms; analyzing posts from different time periods to see consistency; using multiple analytical approaches (content analysis + sentiment analysis + linguistic analysis); comparing with existing survey data on climate anxiety; Cross-referencing with news events or climate reports during the same timeframe.
Author Response
Introduction:
1) Line 34-35: "the great epidemic of the 21st century" needs a reference
We have added an appropriate citation to support the phrase “the great epidemic of the 21st century” in reference to the global mental health crisis. The statement now cites Vázquez Canales (2023), a source already included in the reference list.
2) Line 36: Reword the sentence:…a decrease in quality of life and health life expectancy due to disability…..
We have revised the sentence for clarity and conciseness. The phrase now reads: “a reduction in both quality of life and healthy life expectancy due to disability.”
3) Line 46-47: add reference
We have added a supporting reference to this statement. The revised sentence now cites Berry, Bowen, and Kjellstrom (2010), who provide a causal framework linking climate change and extreme weather to a range of mental health outcomes, including anxiety, PTSD, and depression.
4) Line 48-49: Add ref
We have added a reference to support the claim linking the water crisis, biodiversity loss, and environmental degradation to increased disease risk and food insecurity. The revised sentence now cites McMichael et al. (2006).
5) Line 95-96: Add ref
We have added a reference to support the statement about Conceptual Metaphor Theory. The revised sentence now cites Lakoff and Johnson (1980), who first developed the theoretical framework for understanding abstract thinking through metaphor.
Material and Methods:
6) The methods section employs an appropriate Boolean search strategy and content analysis methodology, but both require more detailed description to ensure replicability. The current approach would benefit from expanded explanations of the search parameters and analytical framework, along with additional information about data management procedures and ethical considerations to meet contemporary research standards.
We have expanded the Methods section to provide a more detailed and transparent description of the study design. Specifically, we now include:
- A full explanation of the Boolean search strategy, including search terms, language filters, platforms, and date range.
- Clarification of the analytical framework, specifying that content analysis was inductive, with emergent categories developed through a grounded coding process.
- A description of data management procedures, including how data were extracted, coded, and stored.
- An explicit statement regarding ethical considerations, including justification for not requiring IRB approval given the use of publicly available content.
7) Overall, I would have liked to see quotes and examples, consolidating your findings.
We have revised the Results section to include representative quotes for each of the four thematic categories. These excerpts help illustrate the discursive patterns identified in our analysis and strengthen the connection between the categories and the underlying data.
8) Line 154: "boolean code" should be "Boolean code"
We have corrected the capitalization of “Boolean code” at Line 154
9) Line 167-168: Are these themes identified and defined by others, in which case you need a reference. If they are identified in this study, should they not be ion the results section!
The four thematic categories—Environmental Awareness Activism, Testimony and Emotional Expression, Scientific and Technical Discourse, and Political–Ideological Positioning—were not pre-established or drawn from existing frameworks. They emerged inductively from the content through a grounded coding process. We have clarified this in the Methods section and moved the list of final categories to the beginning of the Results section, where they are now defined and exemplified.
10) Line 192-197: this is introduced a bit abruptly and would benefit from further explanation. I would recommend generating a table with the complete search strategy; include all constructs use to enable reproducibility.
We have revised the paragraph to improve the transition and provide a clearer introduction to the search process. Additionally, we have added a table summarizing the complete Boolean search strategy, including all constructs and criteria used for inclusion and exclusion. This enhancement improves transparency and supports reproducibility.
11) Line 211-225: How were these five categories identified? Was your analysis underpinned by the coding framework from the ECODES Observatory report?
The five discursive strategies (emotionalization, dramatization, activation, polarization, and legitimation) were adapted from the analytical framework proposed by the ECODES Observatory report. However, we adjusted and applied them inductively to the specific corpus of our study. We have clarified this connection in the manuscript and specified the role of the ECODES framework in shaping the coding process.
Results:
12) The results section presents clear categorization of findings but lacks sufficient depth to fully convey the richness of the data. While the identification of key themes provides a useful framework, the analysis would be strengthened by incorporating representative quotes or examples that illustrate these categories in practice. The notable finding that 41% of participants engaged in environmental activism represents a particularly interesting result that warrants deeper exploration and contextualization within the broader literature on climate engagement behaviors.
We have enriched the Results section by incorporating representative quotes that illustrate each of the four thematic categories. These excerpts provide greater depth and allow readers to better grasp the emotional and rhetorical dynamics of the discourse analyzed. In addition, we have expanded the discussion of the Environmental Awareness Activism category—which comprised 41% of the sample—by contextualizing this finding within the broader literature on climate engagement behaviors and eco-citizenship.
13) Line 339-355: I would recommend moving this section and associated figures to the beginning of the results. I also recommend producing a Word cloud in English and present both.
We have moved the section containing the word frequency analysis and associated figures to the beginning of the Results section, as recommended. To complement the original word cloud in Spanish, we have also generated an English version to facilitate accessibility for international readers. Both visualizations are now presented side by side for comparative purposes.
14) I recommend starting with the quantitative findings would provide readers with the overall landscape and key statistics first, creating a clear foundation before diving into the more nuanced qualitative analysis. This structure would guide readers from the "what" (the numbers and patterns) to the "how and why" (the deeper meanings and explanations), making the results much easier to follow and understand.
We have reorganized the Results section to begin with the quantitative overview. This includes key statistics, distribution of categories, and word frequency analysis.
15) Table 1: Consider adding total sample sizes for context
We have clarified this point by adding a footnote to Table 2, specifying that the total sample consists of 120 digital publications. The note also explains that multiple lexical labels may appear within a single publication, providing context for the reported frequencies.
Discussion:
16) The discussion demonstrates good integration with existing literature and valuable comparisons with other research findings that strengthen the interpretation of results. While the section effectively situates the findings within the broader academic context, it would benefit from a more thorough examination of study limitations and clearer articulation of future research directions to provide readers with a complete understanding of the research scope and potential avenues for advancement.
We have expanded the Discussion to include a more detailed reflection on the study’s limitations, such as the exclusive focus on Spanish-language media, potential selection bias in digital platforms, and the lack of intercoder reliability assessment
17) Line 400: I would have thought qualitative studies were needed, to better understand the nuances. Or perhaps even better, mixed methods.
We fully agree that qualitative and mixed-methods approaches are essential to deepen the understanding of the emotional, linguistic, and sociocultural nuances associated with eco-anxiety and solastalgia. We have revised the Discussion accordingly to emphasize the value of such approaches for future studies.
18) Sample Size and Representativeness. 120 posts is relatively small for digital media content analysis
While 120 publications may appear limited, our sampling strategy prioritized relevance and conceptual richness over volume, focusing specifically on posts that explicitly referenced eco-anxiety or solastalgia within a mental health or emotional context. This allowed us to conduct an in-depth thematic analysis based on clearly delimited inclusion criteria. We have added a brief clarification in the Methods section to explain this sampling rationale.
19) No information about data saturation or sample adequacy
While we did not formally assess data saturation, our approach was not based on theoretical sampling but rather on a bounded corpus of digital media content that met specific inclusion criteria (explicit references to eco-anxiety or solastalgia in Spanish-language publications). Given the exploratory nature of this study and the novelty of the topic in Latin American media discourse, our aim was to identify emerging patterns rather than to reach saturation in a traditional qualitative sense. We have clarified this point in the Methods section to better reflect our rationale.
20) Unclear how representative the sample is of broader Spanish-speaking digital discourse
Selection Bias
Our sampling strategy was not intended to provide a comprehensive view of the entire Spanish-speaking digital discourse, but rather to explore how eco-anxiety and solastalgia are framed in curated digital publications from journalistic and informative sources. The selection focused on textual content from online news and digital press, excluding user-generated or algorithm-driven content such as social media. While this introduces a degree of selection bias, it allowed us to focus on structured, edited discourse that reflects how these terms are constructed and circulated in formal media narratives. We have added a note on this limitation in the manuscript.
21) Manual initial selection process may introduce researcher bias
We recognize the potential for researcher bias in manual selection processes. To mitigate this, inclusion criteria were clearly defined in advance, focusing exclusively on publications that explicitly mentioned eco-anxiety or solastalgia in relation to emotional or mental health impacts. While the selection was conducted manually, decisions were guided by these objective criteria rather than interpretive judgments. We have added a clarification in the Methods section to reflect this.
22) No inter-rater reliability reported for categorization
While we did not calculate a formal inter-rater reliability coefficient, the coding process involved multiple researchers to enhance rigor and reduce individual bias. Two researchers conducted the initial lexical screening and proposed the preliminary categories. These categories were then used by four additional researchers to classify the full set of publications. Discrepancies were discussed and resolved collectively, ensuring consistency and agreement across coders. We have added this information in the Methods section to clarify our approach.
23) Lack of detail about inclusion/exclusion criteria beyond Boolean searches
We have now clarified in the Methods section that only publications explicitly addressing eco-anxiety or solastalgia in relation to emotional or mental health were included. Content focused exclusively on political or economic aspects, scientific announcements without psychological framing, or brief mentions without thematic development were excluded. These refinements were added to improve clarity and replicability.
24) Geographic Representation: While focusing on Spanish-language content, no breakdown by country/region
25) Spanish-speaking communities are diverse; findings may not generalize across all contexts
Due to the nature of the digital sources included—many of which are transnational platforms or syndicated news services—it was not always possible to reliably identify the country of origin of each publication. For this reason, no systematic geographic breakdown was attempted. We acknowledge this as a limitation and have now added a note in the Discussion to clarify this point and suggest that future studies could more explicitly incorporate regional differentiation.
26) Limited Methodological Detail: LIWC analysis mentioned but results not fully presented
We acknowledge that the mention of LIWC may have created the expectation of a formal psycholinguistic analysis. In this study, LIWC was used solely as an exploratory tool during the initial lexical screening phase, to help detect posts rich in emotional or affective content. It was not intended as a central analytic component, and no statistical or category-level results from LIWC were used in the final analysis. We have clarified this in the revised Methods section and removed any ambiguous references that might suggest otherwise.
27) No discussion of analytical framework beyond content analysis
he analytical approach used in this study aligns with an inductive qualitative content analysis framework, guided by the principles of thematic categorization. Although not grounded in a specific theory such as grounded theory or discourse analysis, the analysis followed an iterative process of open coding, category development, and cross-validation among multiple researchers. We have now expanded the Methods section to clarify this analytic framework and to distinguish the descriptive lexical phase from the thematic interpretation phase.
28) Missing information about coding procedures and reliability checks
This point has been addressed. We have clarified in the revised Methods section that multiple researchers were involved in the coding process, and discrepancies were resolved collaboratively. Although no formal inter-rater reliability statistic was calculated, reliability was ensured through consensus-based discussion and shared application of the coding framework.
29) Quantitative Presentation: Percentages provided without confidence intervals or statistical significance testing
Confidence Interval are now provided in table 2.
30) No discussion of whether differences between categories are meaningful
The differences in category frequency (e.g., Environmental Awareness Activism: 41%; Catastrophic Thinking: 25%) are reported to provide readers with a general sense of thematic salience within the analyzed content. However, we agree that these proportions should not be interpreted as statistically meaningful comparisons, given the non-probabilistic nature of the sample. We have clarified this in the revised manuscript, emphasizing that the analysis is exploratory and descriptive rather than comparative or inferential.
31) Validation and Reliability: No member checking or participant validation (If possible. I appreciate it may not be a realistic expectation for this type of digital content analysis but still a limitation worth noting)
We agree with the reviewer that member checking or participant validation was not feasible in this study, given that the data consisted of publicly available digital content from anonymous or uncontactable users. We acknowledge this as a limitation inherent to digital media research and have now added a note in the Discussion section to reflect this constraint.
32) Limited discussion of researcher reflexivity
Thank you for highlighting this important point. We have now added a brief reflexivity statement to the Discussion section, acknowledging how the backgrounds and perspectives of the research team may have influenced the thematic interpretation. The interdisciplinary composition of the team—comprising researchers from psychology, linguistics, and psychiatry—helped to balance potential biases, and coding decisions were regularly discussed to promote interpretive transparency. This is now included as part of the methodological reflection.
33) No triangulation with other data sources or methods: e.g. comparing findings across different social media platforms; analyzing posts from different time periods to see consistency; using multiple analytical approaches (content analysis + sentiment analysis + linguistic analysis); comparing with existing survey data on climate anxiety; Cross-referencing with news events or climate reports during the same timeframe.
As an exploratory study, our focus was on identifying thematic patterns within a delimited corpus of Spanish-language digital publications that explicitly mentioned eco-anxiety or solastalgia during a specific time window. While triangulation with other sources (e.g., different platforms, longitudinal data, or complementary surveys) was beyond the scope of this initial analysis, we agree that such strategies would significantly enrich future research. We have added this reflection to the revised Discussion section, noting triangulation as a key direction for follow-up studies.
Round 2
Reviewer 2 Report
Comments and Suggestions for Authors
Overall, I am content with the revisions that the authors made in response to my comments. This is a fine piece of work; I congratulate the authors.
There is one line in the paper that I do not understand: "By focusing on curated digital publications from online news and press sources, we prioritized the analysis of structured and editorial content over algorithm-driven or user-generated data." (Lines 161-2). Perhaps I misunderstand the types of publications used, but Instagram, Facebook, etc. seem like they would fall under the category of "user-generated data," unless perhaps the authors only looked at posts made by a news media source. I would urge the authors to clarify this statement.
Author Response
We acknowledge that the original sentence was misleading. As is evident throughout the manuscript, we analyzed opinions from citizens (the largest subgroup in our corpus) as well as activists. We have corrected the sentence to reflect that both editorial content and user-generated posts (from citizens and activists) were included:
"Our corpus included both editorially curated publications from established news and press outlets (accessed via their websites or verified social media accounts) and public posts authored by activists and citizens."